# A DC-DC Center-Tapped Resonant Dual-Active Bridge with Two Modulation Techniques

**Gengxin Chen** [1] , **Nuoman Xu** [1], **Li Yuan** [2], **Muhammad Humayun** [1] **and Muhammad Mansoor Khan** [1,*]

[1]  Key Laboratory of Control of Power Transmission and Conversion of Ministry of Education, Shanghai Jiao Tong University, Shanghai 200240, China; whychengengxin@sjtu.edu.cn (G.C.); xu_nuoman@163.com (N.X.); mhumayun88@gmail.com (M.H.)
[2]  State Grid Changzhou Power Supply Branch Company of Jiangsu Power Electronic Co., Ltd., Changzhou 213000, China; wjgdgsyl@sohu.com
[*]  Correspondence: mansoor@sjtu.edu.cn

**Abstract:** Power converters with higher efficiency in a wide load range are important for reducing the overall energy consumption of renewable energy generation systems. A center-tapped LC series resonant dual-active bridge (LC-DAB) converter for DC-DC conversion is proposed in this paper. The proposed converter utilizes a center-tapped bridge to block reverse current and eliminate back flow power to reduce conduction losses. Two modulation methods for the proposed topology (i.e., fixed frequency modulation (FFM), and variable frequency modulation (VFM)) are proposed and analyzed. Both modulation methods can realize soft switch over the entire load range to reduce switching losses. In addition, the proposed modulation techniques guarantee soft switching for all devices and synchronous rectifier is realized by the center-tapped bridge to further reduce the conduction losses. Furthermore, a comprehensive comparison in terms of conduction losses and switching losses has been carried out to highlight the superiority of the proposed converter over the existing LC resonant converters. Finally, simulated and experimental results for a 1.5 kW prototype are presented to validate the theoretical analysis and performance of the proposed converter.

**Keywords:** LC series resonant; dual-active bridge; fixed frequency modulation; variable frequency modulation; border conduction mode

## 1. Introduction

A dual-active bridge (DAB) converter can be considered as a regulated DC transformer, capable of adjusting its output voltage under varying load and input voltage conditions [1–4]. Due to its potential in high power density, soft switching capability, galvanic isolation and flexible modulation, DAB has become an attractive candidate for applications such as renewable energy harvesting, new energy vehicle chargers and DC micro-grid distribution networks [5,6].

The power conversion of DAB is realized by a full-bridge converter cascaded by an inductive tank and a cyclo-converter through a high-frequency transformer (HFT). Maximization of the DAB converter's effects with a minimum number of components is a major concern when designing new bipolar converters. In this context, some new bipolar DAB converters have been proposed through the use of bidirectional switches in [7,8]. Compared to conventional unidirectional MOSFETs, bidirectional switches can block voltage from both directions and therefore ensure unidirectional power flow. In order to further reduce the number of switching devices and achieve bipolar voltage without auxiliary circuits, Sun et al. developed the idea of a center-tapped bridge [9]. Although these topologies have the potential to reduce the conduction losses by eliminating the back-flow current,

these converters suffer from higher conduction losses, as they linearly increase current through the inductor, which limits the performance and power density.

In practice, DAB-coupled resonant tanks are considered as one of the most practical available DAB converters [10–15]. LC resonance is introduced in [10,11] for three-phase and single-phase converters. Twiname et al. proposed a resonant converter with an LCL and CLC tank in [12,13], both of which can reduce conduction loss by introducing resonance in the current. In [14], Mansouri et al. proposed a CLLC resonant tank, which can also reduce conduction losses. Shakib and Mekhilef proposed new LLC resonant converter structures and extended the input voltage range [15]. These converters had a higher attraction for researchers due to their low conduction losses and high power density. Compared to other resonant tank circuits, the LC resonant tank can achieve zero impedance at the resonant frequency. Therefore, LC-DAB can increase the power density with reduced number of components [16]. Furthermore, the resonant capacitor can also play an important role as a DC-blocking capacitor, which protects the inductor and transformer from saturation. Despite of all their exciting benefits, the traditional LC resonant converters require heavy and bulky passive components which leads to an increased system volume and cost. To address these shortcomings, significant efforts have been made by utilizing a higher switching frequency. The sizes of the passive components are noticeably reduced, but the switching losses are substantially increased.

In order to enhance the efficiency through reduced switching losses, new modulation techniques based on fixed switching frequency are presented to realize soft switching in [17–20]. The fixed frequency modulation (FFM) is simple to design and can reduce the control complexity. Beside the considerable reduction in switching losses through soft switching, some partial advantages related to design, heat sink, and reliability are directly obtained. Hence, a duty ratio control method is proposed in [17] in order to realize soft switching under heavy load and keep the output voltage constant under different loads. However, the soft-switching range of this converter is narrow and the maximum efficiency is obtained at the peak voltage gain, which is not suitable for wider load range applications (e.g., energy storage battery charging). Wide-range soft-switching modulations are designed to solve the light-load ZVS turning-on issue. Safaee et al. [18] analyzed the resonant characteristics of LC-DAB in the time domain, and obtained the current at the switching time point, which helps in designing parameters for soft switching modulations. A phase shift control strategy has been proposed [20] to realize ZVS to achieve soft switching in a wider current bandwidth, but the converter operates in a narrow output voltage bandwidth. Although FFM has several benefits, it has numerous shortcomings (i.e., it cannot optimize the current waveform in a wide load range).

Therefore, a variable frequency modulation (VFM) technique is proposed by Shakib and Mekhilef in order to solve the wider voltage range issue (i.e., to modulate the impedance of the LC resonant tank by adjusting the switching frequency). The proposed VFM [21] can not only extend the voltage range, but it also helps realize soft switching. However, they did not consider harmonic components in the harmonic current, which can influence the accuracy of the estimated power. Ping et al. [22] considered harmonics in their converter. Nevertheless, the problem of back-flow power and high switching current at low power operation still remains due to its single-phase shift modulation. Optimizing the harmonic effects and minimizing the losses are two major challenges in the development of new modulation techniques.

This paper investigates a novel LC series resonant DAB and two novel modulation techniques based on fixed and variable switching frequencies for the proposed converter. Through the proposed modulation scheme, switches at the primary side and the secondary side of the HFT are commutated softly without any auxiliary circuits. Hence, the modulation techniques reduce the converter power losses for the entire load range without extra costs. Therefore, the main contribution of this paper is summarized as follows:

1. A novel topology of LC series resonant DAB is proposed. The input side full-bridge converter is cascaded by the output side center-tapped bridge through a center-tapped high-frequency transformer (CT-HFT) associated with an LC series resonant tank. The CT-HFT can provide a

galvanic isolation between the input and output link. The center-tapped full bridge network comprises two pairs of anti-series MOSFETs and hence can block reverse current;

2. Two modulation methods, FFM and VFM, are developed and tested. The efficiency of the proposed converter is significantly improved (96.5%) by applying the FFM technique in a discontinuous conduction mode (DCM) that inherits the switching loss reduction and eliminates the back-flow current. At heavier load, the conduction losses are significantly reduced by 40% through the VFM technique in border conduction mode (BCM), and the system efficiency is improved (up to 97.6%). The proposed modulation technique increases the power transfer capabilities and broadens the voltage gain, under which the converter still maintains soft switching without adding lossy components or complicated modulation. As a result, higher conversion efficiency and reliability are achieved;

3. The current waveforms under FFM and VFM are accurately described and analyzed in time domain;

4. The detailed analysis and comparison between the proposed modulation technique with the conventional LC-DAB converter, presented in [17], are carried out. The aspects of comparison include conduction losses, switching losses and converter efficiencies;

5. Finally, the simulated and experimental results are presented and the efficiencies are compared.

This paper is further organized as follows. The topology and the two modulation algorithms of the proposed LC-DAB are introduced in Section 2. The control algorithm is given in Section 3. In Section 4, the loss and power efficiencies of the conventional converter and the proposed converter under FFM and VFM are calculated theoretically and compared in terms of switching losses, conduction losses and total losses. Finally, the experimental results and a conclusion are presented in Sections 5 and 6.

## 2. The Proposed Converter

Figure 1 shows the circuit architecture of the LC-DAB DC-DC converter. The power conversion is realized by input side full bridge network and output side center-tapped bridge. The full bridge consists of four MOSFETs $S_1 - S_4$ with antiparallel diodes $D_1 - D_4$. The center-tapped bridge comprises two four-quadrant switches, both of which are realized by a pair of MOSFETs $S_{P1}$, $S_{N1}$ and $S_{P2}$, $S_{N2}$ with antiparallel body diodes $D_{P1}$, $D_{N1}$ and $D_{P2}$, $D_{N2}$ connected in anti-series fashion. The current in reverse direction can flow through the body diode of the MOSFET in a full bridge and leads to backflow power. In comparison, the center-tapped bridge is capable of blocking the flow of current in the reverse direction. When both MOSFETs in a leg of the center-tapped bridge are off (e.g., $S_{P2}$ and $S_{N2}$), then no current can flow through that leg. When one of the MOSFETs is switched on (e.g., $S_{P1}$), only the forward current (positive $i_r$) can flow through that leg and the reverse current (negative $i_r$) will be blocked by the other MOSFET (i.e., $S_{N1}$). When reversing the gate signals for this leg, the current flow direction will also be reversed. Therefore, we can limit the current flow direction by modulating the gate signals. The primary side the DC bus voltage is $U_1$, and $U_2$, which stands for the output side voltage. $C_{IN}$ and $C_{OUT}$ are filter capacitors placed near the bridges to improve electromagnetic compatibility. The resonant inductor L and capacitor C are connected to the CT-HFT in series to increase power density.

The input side bridge works as a classical full bridge, and the output voltage of this bridge is denoted by $u_1$. $u_1$ is applied across the LC resonant tank and the CT-HFT. The secondary side voltages of CT-HFT are denoted by $u_2$ and $u_3$, which are equal in magnitude and phase. If $u_2 > 0$, the devices in upper leg of the center-tapped bridge will conduct and vice versa. The conduction of MOSFETs and diodes depends on the direction of the current. As mentioned in the introduction section, the center-tapped bridge can generate a bipolar voltage at the output side, which is one of its main advantages. The control of these switches is comparatively complex and new proper modulation techniques are needed to ensure soft switch and high efficiency.

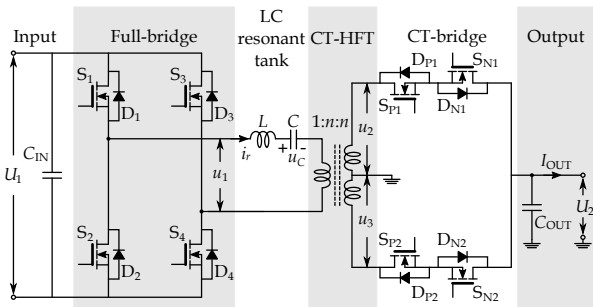

**Figure 1.** Proposed Converter Schematic.

Table 1 compares the switching performance and conducting performance between conventional isolated DC-DC topologies and the proposed center-tapped topology (CT-LC-DAB). The conventional center-tapped L-DAB (CT-L-DAB) consists of one full bridge and a center-tapped bridge, an inductor and a CT-HFT in between [23]. The output voltages of the two bridges are rectangular waves, and the phase shift between the square waves controls the power flow. However, this topology can not avoid back-flow power, which increases the conduction loss. The conventional LC-DAB (FB-LC-DAB) is made up of two full bridges, an LC resonant network and an HFT. The output voltage of the output side is square wave and the input side bridge generates a rectangular wave, which is center-aligned with the square wave. The power flow is controlled by the duty ratio of the rectangular wave. Nevertheless, the conventional LC-DAB causes high switching loss due to hard switching.

**Table 1.** Comparison between proposed topology and conventional topology.

| Topology | Hard Switching | Back Flow Current |
|----------|:---------------:|:------------------:|
| CT-L-DAB [9] | No | Yes |
| FB-LC-DAB [20] | Yes | No |
| CT-LC-DAB | No | No |

The modulation signal waveforms and conduction loops of the proposed FFM and VFM are introduced, and a detailed mathematical analysis will be given in the following subsections. The analysis in this section is carried out under the assumption of ideal components, in which the parasitic parameters, internal resistance and switching process are ignored.

*2.1. Fixed Frequency Modulation (FFM)*

Figure 2 shows the operation waveforms of the proposed converter under fixed frequency modulation. The switching frequency under FFM is fixed at the resonant frequency given in Equation (1).

$$f_{\text{FFM}} = \frac{\omega_r}{2\pi} = \frac{1}{2\pi\sqrt{LC}} \tag{1}$$

$$T_{\text{FFM}} = \frac{1}{f_{\text{FFM}}} \tag{2}$$

The amplitude of the output current is controlled by the phase shift in the primary side. $S_1$ and $S_2$, $S_3$ and $S_4$ are always switched in complement. The phase shift ratio of the switch signal between $S_1$ and $S_3$ is $1 - d$, where $d$ denotes the duty ratio of bridge voltage $u_1$. Each operation sequence can be divided into six stages. Stage 1–3 make up half a sequence, namely $t_3 = T_{\text{FFM}}/2$, $t_6 = T_{\text{FFM}}$. The time intervals of Stage 1 and Stage 4 are the same, so are Stage 2 and Stage 5. So, $t_4 = t_1 + T_{\text{FFM}}/2$ and $t_5 = t_2 + T_{\text{FFM}}/2$.

In Stage 1 and Stage 2, the inductor current is positive, hence the capacitor voltage increases. Similarly in Stage 4 and Stage 5, the capacitor voltage decreases because of the negative inductor current. In Stage 3 and Stage 6, the voltage across the resonant capacitor remains unchanged due to the zero inductor current.

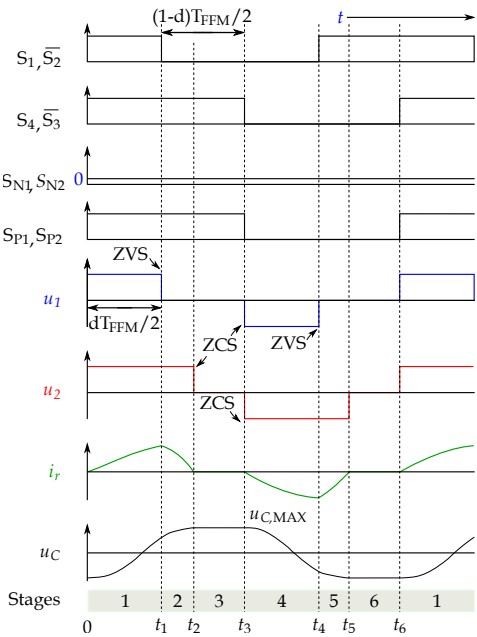

**Figure 2.** Key waveform of the proposed converter with FFM.

For simplification, the output side voltage is transferred to the primary side by dividing the turns ratio.

$$U_2' = \frac{U_2}{n} \tag{3}$$

In the positive operation condition, $U_1$ and $U_2'$ are positive and $U_1 > U_2'$. If the operation condition $U_1 > U_2'$ does not hold, there will be no power delivered from the DC side to the AC side. In the meantime, the switches $S_{N1}$ and $S_{N2}$ will prevent the current flowing reversely. As a result, the power conversion stops until $U_2'$ becomes lower than $U_1$. Assume that the input and output DC link capacitance is large enough, then $U_1$ and $U_2$ are constant during the operation. The conduction loops for positive output voltage in each stage are illustrated in Figure 3.

1.　Stage 1 (Figure 3a) ($0 < t < t_1$): The initial inductor current is zero and the capacitor voltage starts from the negative peak value $U_{C,MAX}$. $S_1$, $S_4$ and $S_{P1}$ are turned on (ZCS), and other switches remain off. The voltage across the LC resonant tank in this stage is $U_1 - U_2'$. The conduction loop can be considered as an LC circuit with constant voltage source and initial capacitor voltage. Therefore, the inductor current and the capacitor voltage changes sinusoidally with amplitude parameter $A_1$, initial phase $\varphi_1$ and voltage bias $B_1$, as shown in Equation (4). The voltage bias is equal to the voltage of the source Equation (6). $\omega_r$ denotes the resonant angle frequency in Equation (1). The value of the amplitude and the initial phase will be discussed after introducing all the stages.

$$i_r(t) = A_1\omega_r C sin(\omega_r t + \varphi_1) \tag{4}$$

$$u_C(t) = A_1 sin(\omega_r t + \varphi_1) + B_1 \tag{5}$$

$$B_1 = U_1 - U_2' \tag{6}$$

2. Stage 2 (Figure 3b) ($t_1 < t < t_2$): At $t_1$, $S_1$ switches off, and $S_2$ turns on with a negative current, namely ZVS. The switches remain unchanged on the output side. The voltage across the resonant tank is negative $U_2'$. The resonant current decreases sinusoidally with amplitude parameter $A_2$ and phase $\varphi_2$, as given in Equation (7). The capacitor voltage, which reaches its positive peak value $U_{C,MAX}$ at $t_2$, can be described by Equation (8) with bias $B_2$.

$$i_r(t) = A_2\omega_r C sin(\omega_r(t - t_1) + \varphi_2) \tag{7}$$

$$u_C(t) = A_2 sin(\omega_r(t - t_1) + \varphi_2) + B_2 \tag{8}$$

$$B_2 = -U_2' \tag{9}$$

3. Stage 3 (Figure 3c) ($t_2 < t < t_3$): The current through the LC tank falls to zero, and thus $D_{N1}$ turns off naturally at $t_2$. The MOSFET $S_{N1}$ will prevent the reverse current flows. The magnitude of the voltage across $S_{N1}$ is $U_2 + nU_{C,MAX}$. This stage is a zero-current stage, which guarantees that the current flows to the output voltage link unidirectionally and the next stage starts with initial current set at zero. In Stage 3, there is no variation in current and the voltage of each component;

4. Stage 4 (Figure 3d) ($t_3 < t < t_4$): At the beginning of Stage 4, $S_2$, $S_3$ and $S_{P2}$ are switched on with zero current (ZCS). The voltage across the resonant tank is $U_2' - U_1$. The inductor current decreases from zero, and the capacitor voltage reduces from the positive peak value. The value of the inductor current and the capacitor voltage are opposite to that in Stage 1;

5. Stage 5 (Figure 3e) ($t_4 < t < t_5$): $S_1$ is turned on with reverse current (ZVS) and $S_2$ is off at $t_4$. The voltage across the resonant tank is $U_2'$. The inductor current rises back to zero and the capacitor voltage reaches negative $U_{C,MAX}$ at $t_5$. The current and voltage values in Stage 5 and 2 are opposite;

6. Stage 6 (Figure 3f) ($t_5 < t < t_6$): At $t_5$ the inductor current is zero and $D_{P2}$ turns naturally off. Similar to Stage 3, $S_{N2}$ blocks the negative voltage $U_2 + nU_{C,MAX}$. Stage 6 lasts until the next sequence starts at $T_{FFM}$.

The modulation duty ratio is $d_{FFM} = 2t_1/T_{FFM}$. Due to the continuity of the inductor current and capacitor voltage, the boundary conditions at the end of Stage 1 are $t_1^-$ and at the beginning of Stage 2 ($t_1^+$).

$$i_r(t_1^-) = i_r(t_1^+) \tag{10}$$

$$U_C(t_1^-) = U_C(t_1^+) \tag{11}$$

As introduced, Stage 1–3 make up half a cycle, and Stage 3 is a zero-current stage. Hence, the relationship of the parameters at $t = 0$ and $t = T_{FFM}/2$ is given as

$$i_r(0) = i_r(t_2) = i_r(T_{FFM}/2) = 0 \tag{12}$$

$$U_C(0) = -U_C(t_2) = -U_C(T_{FFM}/2) \tag{13}$$

The amplitudes and the initial phase angles can be derived by solving Equations (4)–(13).

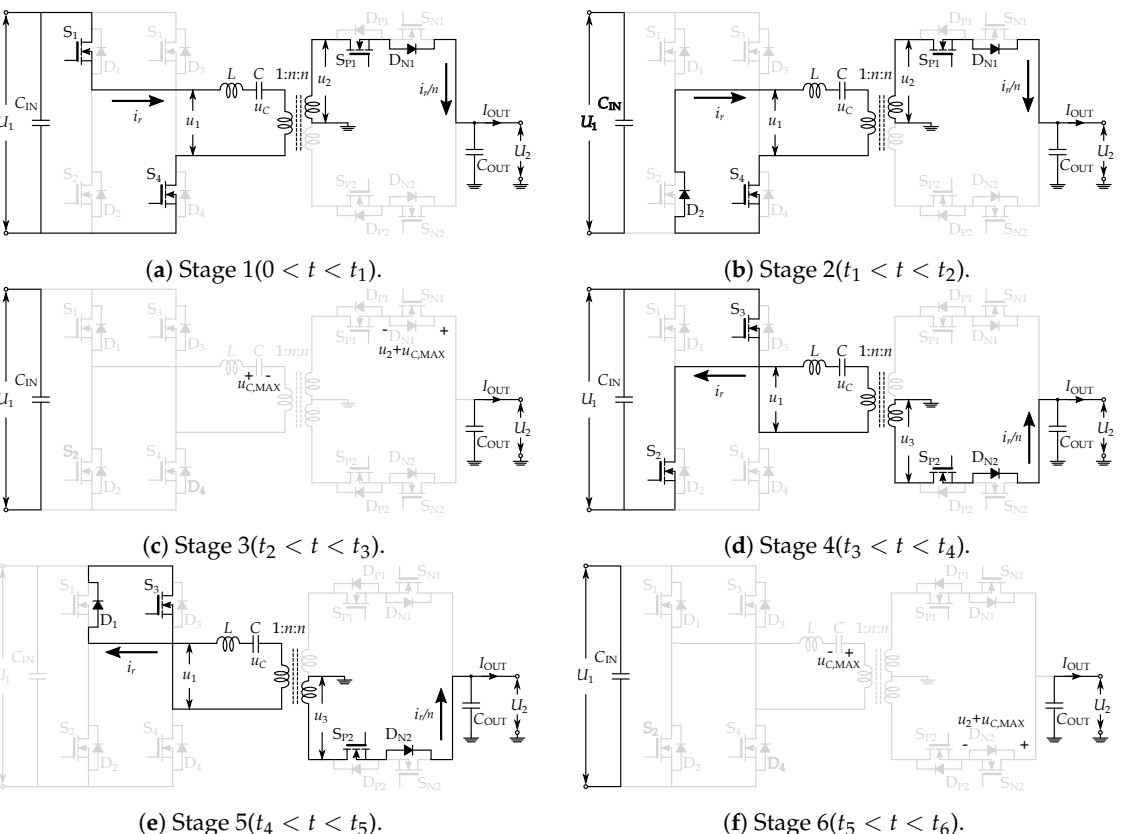

(**a**) Stage 1($0 < t < t_1$).

(**b**) Stage 2($t_1 < t < t_2$).

(**c**) Stage 3($t_2 < t < t_3$).

(**d**) Stage 4($t_3 < t < t_4$).

(**e**) Stage 5($t_4 < t < t_5$).

(**f**) Stage 6($t_5 < t < t_6$).

**Figure 3.** Operation stages of the proposed converter ($U_2 > 0$).

$$A_1 = \frac{2U_2'(U_1 - U_2')}{2U_2' - U_1 + U_1 cos(\omega_r t_1)} \tag{14}$$

$$A_2 = A_1 - U_1 + 2U_2' \tag{15}$$

$$\varphi_1 = -\pi/2 \tag{16}$$

$$\varphi_2 = arccos\left(\frac{A_1}{A_2}sin(\omega_r t_1)\right) \tag{17}$$

And the value of $t_2$ can be obtained by

$$t_2 = \frac{\pi/2 - \varphi_2}{\omega_r} + t_1 \tag{18}$$

The duty ratio of the full bridge is

$$d_{\text{FFM}} = \frac{2t_1}{T_{\text{FFM}}} \tag{19}$$

### 2.2. Variable Frequency Modulation (VFM)

Fixed frequency modulation consists of four power delivering stages and two zero-current stages. The existence of Stage 3 and Stage 6 does not contribute to power transmission, but prolongs the period and results in a higher RMS/average current ratio [24,25]. The RMS current decides the conduction

loss, and average current equals the output current. Variable frequency modulation will shorten the period, and let the duration of Stage 3 and Stage 6 become zero. Consequently, zero-current stages are eliminated and the resonant current rises in the reverse direction right after returning to zero (i.e., in BCM). The advantage of VFM is lower RMS current and hence lower conduction loss.

To achieve BCM, Stage 1 and Stage 4 are supposed to start immediately when Stage 5 and Stage 2 end (when the inductor current returns zero), as shown in Figure 4. The zero-crossing point of the inductor current can be detected by a current sensor. Nevertheless, online detection is costly. An alternative strategy is to predict the zero-crossing point of the inductor current based on the operation conditions. In Section 2.1, the value of $t_2$ is obtained, which is the time when the inductor current reaches zero. In VFM algorithm, $t_2$ marks the half period of the switching cycle. Therefore, the value of the period for VFM algorithmis is given as

$$T_{\text{VFM}} = 2t_2 = \frac{2(\pi/2 - \varphi_2)}{\omega_r} + 2t_1 \tag{20}$$

Hence, the switching frequency and the duty ratio $d$ are given as

$$f_{\text{VFM}} = \frac{1}{T_{\text{VFM}}} \tag{21}$$

$$d_{\text{VFM}} = \frac{t_1}{T_{\text{VFM}}} \tag{22}$$

If the frequency and the duty ratio satisfy the above relationship, BCM operation can be realized. The upper limit of $t_1$ is dependent on the input and output voltages

$$t_{1,\text{MAX}} = \frac{1}{\omega_r} arccos\left(\frac{U_1 - 2U_2'}{U_1}\right) \tag{23}$$

$t_{1,\text{MAX}}$ is also the upper limit of $t_1$ under FFM. The duty cycle $T_{\text{VFM}}$ is positively related to $t_1$. When $t_1 = t_{1,\text{MAX}}$, the cycle period under VFM will be equal to the resonant duty cycle, and the output current will be theoretically infinite.

$$T_{\text{VFM,MAX}} = T_{\text{FFM}} = 2\pi\sqrt{LC} \tag{24}$$

At a given operation voltage $U_1$ and $U_2'$, if $t_1$ is chosen between 0 and $t_{1,\text{MAX}}$, there always exists a corresponding cycle period, $T$. Furthermore, the converter outputs a corresponding cycle average current, which will be derived in Section 3. If $t_1$ goes beyond $t_{1,\text{MAX}}$, Equation (20) is still established and the current reaches zero at $t = T_{\text{VFM}}/2$. However, the peak current in each cycle will increase continuously.

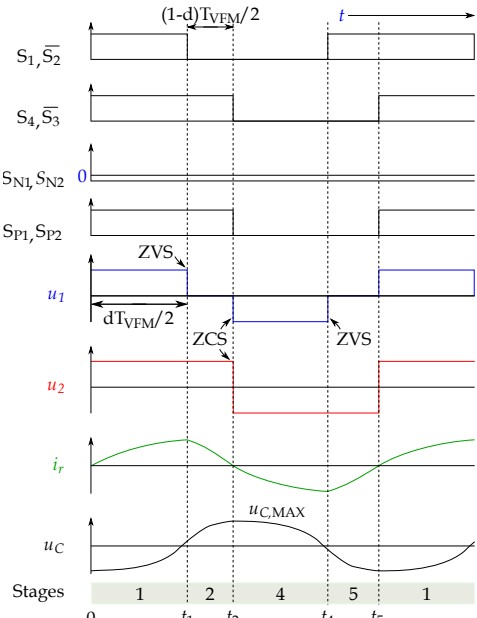

**Figure 4.** Key waveform of the proposed converter with VFM.

## 2.3. Realization of Synchronous Rectifier

To further reduce the conduction loss, some of the off-state MOSFETs in the center-tapped bridge can be turned on in some stages, so that the current flows through the channel of the MOSFETs instead of their anti-parallel diodes. For positive output, $S_{N1}$ is turned on in Stage 1 and Stage 2, and $S_{N2}$ is turned on in Stage 4 and Stage 5. Similarly, for a negative output, $S_{P1}$ and $S_{P2}$ will be turned on in Stage 1 and Stage 2, Stage 4 and Stage 5, respectively. These MOSFET will be alternately ZCS turned on at beginning of each cycle after the insertion of suitable dead time. The turn-off time for these MOSFET should be set slightly earlier, before the end of the stage, to avoid reverse current flow and achieve ZCS turn-off. This modulation method can be applied in both FFM and VFM.

## 3. Control Algorithm

The control methods for the proposed converter can be open-loop and closed-loop. The duty ratios of the switching signals in open-loop are estimated based on the analysis in Section 2. The closed-loop control requires feedback signal and a PI controller to eliminate the estimation error in open loop. The details of open-loop control and closed-loop control are discussed in this section.

### 3.1. Open-Loop Control

The open-loop modulation parameters (period $T$ and phase shift ratio $d$) for this converter can be calculated from the reference output current $I_{OUT}$, the input voltage $U_1$ and the output voltage $U_2$. $U_1$ and $U_2$ are sampled and measured by the micro processor. The calculation process and results are given in this section. The flow chart for calculating the modulation parameters is illustrated in Figure 5.

In half a sequence, the capacitor voltage varies monotonically due to the unidirectional current. Thus the output current (namely the average current) can be derived from the variation in the capacitor voltage.

$$I_{OUT} = \frac{2C}{nT} \left| u_C(0) - u_C(T/2) \right| = \frac{4C}{nT} \left| u_C(0) \right| \tag{25}$$

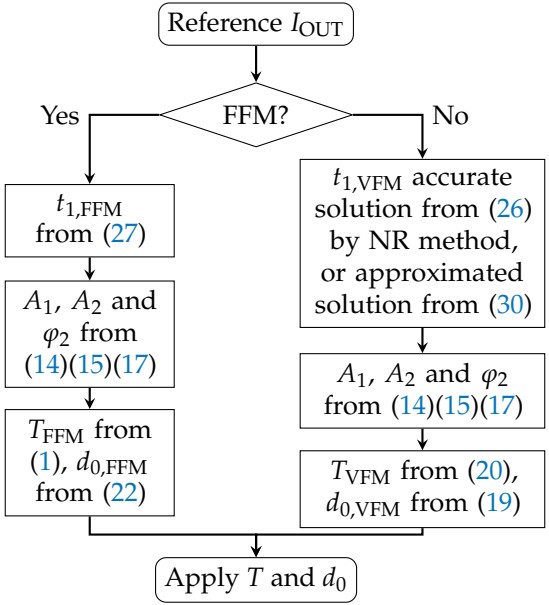

**Figure 5.** Control Flow Chart for the Proposed Converter.

From Equations (5)–(17) and (25), the accurate value of the output current is obtained.

$$I_{\text{OUT}} = \frac{4C}{nT}\left(U_1 - U_2' + \frac{2U_2'(U_1 - U_2')}{2U_2' - U_1 + U_1 cos(\omega_r t_1)}\right) \tag{26}$$

This equation holds for both VFM and FFM, depending on the choice of $T$. For FFM, the period is a constant $T = T_{\text{FFM}}$, and the period for VFM is a varying $T_{\text{VFM}}$. The solution of $t_{1,\text{FFM}}$ can be obtained by Equation (27).

$$t_{1,\text{FFM}} = \frac{1}{\omega_r} arccos\left(\frac{8CU_2'(U_1 - U_2')}{U_1\left(nT_{\text{FFM}}I_{\text{OUT}} + 4C(U_1 - U_2')\right)} + \frac{U_1 - U_2'}{U_1}\right) \tag{27}$$

However, Equation (26) for VFM does not have an analytic solution. This solution can be numerically calculated by the Newton–Raphson (NR) method. For fast online calculation, an approximated solution can be obtained by applying Equations (28) and (29).

$$cos(\omega_r t_1) \approx 1 - (2\omega_r t_1/\pi)^2 \tag{28}$$

$$T_{\text{VFM}} \approx \frac{t_{1,\text{VFM}}U_1}{U_2} \tag{29}$$

The approximated solution of Equation (26) is given by

$$t_{1,\text{VFM}} \approx \frac{-b + \sqrt{b^2 - 4ac}}{2a} \tag{30}$$

With coefficients

$$a = \frac{-nU_1^2\omega_r^2}{CU_2'(U_1 - U_2')} I_{OUT} < 0 \tag{31}$$

$$b = -2U_1\omega_r^2 < 0 \tag{32}$$

$$c = \frac{nU_1\pi^2}{2C(U_1 - U_2')} I_{OUT} > 0 \tag{33}$$

The estimation error is caused by the error in Equations (28) and (29). For the approximation of cosine function, the error depends on the value of $t_1$. In the range of $0 < \omega_r t_1 < \pi/2$, the error is relatively small. When $\omega_r t_1$ is beyond $\pi/2$, the error starts to increase rapidly. For the approximation of $T_{\text{VFM}}$, Equation (20) shows that the error is positively related to $t_1$. Therefore, both parts of the error are dependent on the value of $t_1$, and for this reason, the largest $t_1$ leads to the largest estimation error. From Equations (26), (1) and (20), it can be seen that the value of $t_1$ is positively related to $U_2$, $I_{OUT}$ and $L$ and negatively related to $C$. The parameters $U_2$ and $I_{OUT}$ are determined by the operation point, while $C$ and $L$ are the designed parameters of the converter. Therefore, from the aspect of design, a smaller resonant inductance and a larger resonant capacitance lead to smaller error. To completely eliminate the estimation error, the capacitance is considered to be infinitely large, and the LC-RDAB will finally become a non-resonant L-DAB. The balance point between performance and estimation error shall be found. For the parameters chosen in this paper, the maximum error is 4%, which is still within a reasonable limit.

Besides the estimation error, the internal resistances of the devices (including inductor, capacitor, MOSFETs, transformers and wires) will also impact the waveforms. They will increase the overall impedance of the conduction loop, and hence lower the real output power. Due to the resistances, the current will fall back to zero earlier than expected, which will not affect the ZVS and ZCS soft-switching. Besides, the error caused by the estimation and the conducting resistances will not cause a serious deviation in the output power. On the one hand, these errors are relatively small, and power controllers will be implemented in the converter, which can compensate for the errors.

The actual output current controlled by the estimation value without a controller will unavoidably deviate from the reference value. The estimation errors at different output voltages are illustrated in Figure 6 using the parameters in Table 2. The values are calculated in MATLAB: the real values are calculated using the Newton–Rapson method (accuracy 0.01%), and the estimation values are calculated using the estimation method introduced in the paper.

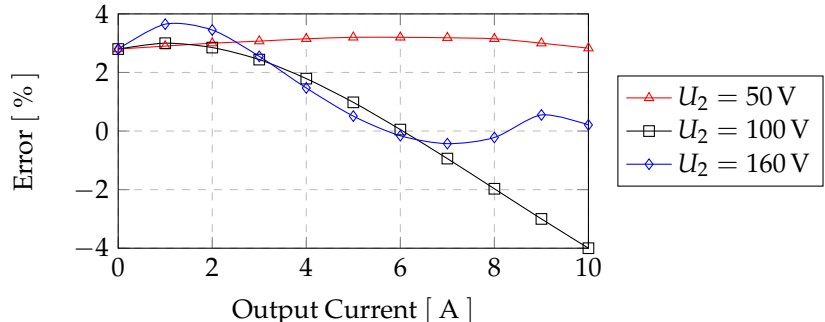

**Figure 6.** Output current error of VFM by approximation. (Parameters in Table 2).

The difference between the actual output current and the reference current in the whole operation range is less than 4% in the designed load range, which suggests that the output current at the approximated frequency and duty ratio can follow the reference value with small error.

*3.2. Close-Loop Control*

For higher control accuracy, a PI controller can be implemented in the closed-loop control to compansate the estimation error and the component parameter deviation. The control loop is shown in Figure 7, where $G_C(s)$ and $G_s(s)$ denote the transfer function of the controller and the converter, respectively. The bias $d_0$ is the estimated duty ratio derived from Figure 5. Based on the analysis in Section 2, the open loop voltage gain $K$ under resistive load changes with the operating point as shown in Equation (34), and the main time constant $\tau$ is dependent on the output capacitance, as illustrated in Equation (35). The switching periods under both FFM and VFM are much shorter than the main time constant, therefore, the overall time constant approximates the main time constant.

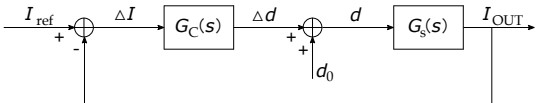

**Figure 7.** Closed-loop control diagram of the proposed converter with PI controller.

$$K = \frac{\Delta I_{\text{OUT}}}{\Delta d} = \frac{2\pi}{n} \frac{U_1 I'_{\text{OUT}} \left( U_1 - U'_2 + I'_{\text{OUT}}/(8\pi C \sqrt{LC}) \right)}{U'_2 \left( 2U'_2 - U_1 - U_1 cos(\omega_r t_1) \right)} \tag{34}$$

$$\tau = n^2 \frac{U'_2 C_{\text{OUT}}}{I'_{\text{OUT}}} \tag{35}$$

The coefficiencies of the PI controller are supposed to be designed according to maximum attainable gain, such that the system can be stable and give a reasonable performance. Therefore, the transfer function of a suitable PI controller can be

$$G_C = K_i \frac{1 + sT_i}{sT_i} \tag{36}$$

with coefficiencies

$$K_i = 1/K_{\text{MAX}} \tag{37}$$

$$T_i = \tau \tag{38}$$

## 4. Power Loss and Efficiency

The power losses consist mainly of conduction losses and switching losses. In this section, a comparative study is carried out between the conventional converter and the proposed converter with FFM and VFM. The topology of the conventional converter is proposed and analyzed in [17,18,20], which consists of two full bridges, an LC resonant tank and a high-frequency transformer in between. Their theoretical losses are compared separately in terms of conduction loss and switching loss. The parameters for the converter are listed in Table 2. The MOSFET parameters ($r_{\text{ON,MOSFET,IN}} = 8$ m$\Omega$ and $r_{\text{ON,MOSFET,OUT}} = 45$ m$\Omega$ for conduction losses, $Q_{\text{gs}} = 54$ nC and $Q_{\text{gd}} = 52$ nC for switching losses [26]) utilize the values given in the datasheet.

**Table 2.** Circuit parameters of the prototype.

| Item | Description | Value |
|------|-------------|-------|
| $U_1$ | Rated Input Voltage | 80 V |
| $U_{2,MAX}$ | Rated Output Voltage | 160 V |
| $P_{MAX}$ | Rated Output Power | 1.5 kW |
| $L$ | Resonant Inductance | 7.5 μH |
| $C$ | Resonant Capacitance | 15 μF |
| $n$ | Turns Ratio | 1:2.2:2.2 |
| $f_r$ | Resonant Frequency | 15 kHz |
| $K_i$ | Controller Coefficiency | 0.078 |
| $T_i$ | Controller Coefficiency | 0.00026 |

### 4.1. Conduction Losses

Conduction losses are decided by the RMS current and the conduction resistance Equations (39) and (40). The resistance in the conduction loop is made up of the on-state resistance $r_{ON,MOSFET}$ of the MOSFETs and the internal resistance of other components $r_{INT}$ Equation (41).

$$P_{COND} = \frac{2}{T} \int_0^{T/2} i_r^2 r_{ON}\, dt = I_{RMS}^2 r_{ON} \tag{39}$$

$$I_{RMS} = \sqrt{\frac{2}{T} \int_0^{T/2} i_r^2\, dt} \tag{40}$$

$$r_{ON} = \left(2 + \frac{2}{n^2}\right) r_{ON,MOSFET} + r_{INT} \tag{41}$$

where $1:n$ is the turns ratio of the transformer. The RMS current of the proposed DAB under FFM is derived as follows. Due to the symmetry of the waveform, the RMS value in steady state can be represented by the RMS value in half cycle, which is calculated in stages in Equations (42)–(44).

1. Stage 1:

$$\int_0^{t_1} i_r^2\, dt = \int_0^{t_1} (A_1 \omega_r C)^2 sin^2(\omega t)\, dt$$
$$= A_1^2 \omega_r C^2 \left(\frac{\omega_r t_1}{2} - \frac{1}{4} sin(2\omega_r t_1)\right) \tag{42}$$

2. Stage 2:

$$\int_{t_1}^{t_2} i_r^2\, dt = \int_{t_1}^{t_2} (A_2 \omega_r C)^2 sin^2(\omega_r(t - t_1) + \varphi_2)\, dt$$
$$= A_2^2 \omega_r C^2 \left(\frac{\pi}{4} - \frac{\varphi_2}{2} - \frac{1}{4} sin(2\varphi_2)\right) \tag{43}$$

3. Stage 3:

$$\int_{t_2}^{T_{FFM}/2} i_r^2\, dt = 0 \tag{44}$$

For each average output current, the parameters used in the above equations are known in Section 2. The VFM method only takes the first two stages in consideration. The overall RMS current under FFM and VFM isdescribed by Equations (45) and (46).

$$I_{\text{RMS,FFM}} = \sqrt{\frac{2}{T_{\text{FFM}}} \left( \int_0^{t_{1,\text{FFM}}} i_r^2 \, dt + \int_{t_{1,\text{FFM}}}^{t_{2,\text{FFM}}} i_r^2 \, dt + 0 \right)} \tag{45}$$

$$I_{\text{RMS,VFM}} = \sqrt{\frac{2}{T_{\text{VFM}}} \left( \int_0^{t_{1,\text{VFM}}} i_r^2 \, dt + \int_{t_{1,\text{VFM}}}^{t_{2,\text{VFM}}} i_r^2 \, dt \right)} \tag{46}$$

Note that FFM and VFM require different $t_1$ and $t_2$ at the same output current, thus their RMS currents are also different. In order to analyze the current of the conventional converter, fundamental harmonic analysis (FHA) is applied [17]. The RMS current of the conventional converter can be approximately obtained as

$$I_{\text{RMS,CONV}} = \frac{n\pi}{\sqrt{8}} I_{\text{OUT,CONV}} \tag{47}$$

Figure 8 shows the conduction losses of three converters at different values of output current.

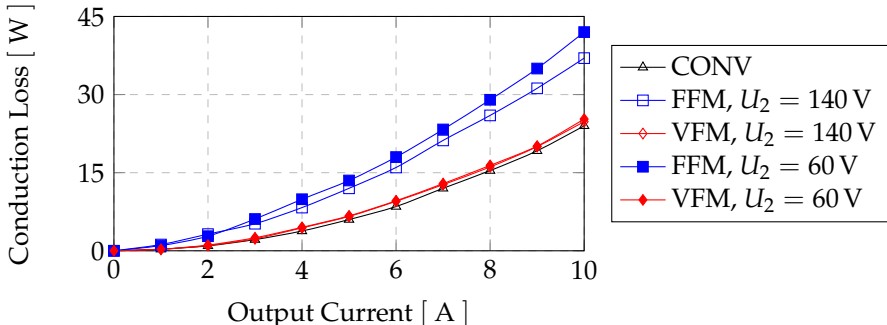

**Figure 8.** Conduction loss of three converters ($U_1$ = 80 V, $r_{\text{ON}}$ = 40 mΩ).

The input and output voltages can influence the RMS current by deciding $t_1$ and $t_2$. The proposed converter under FFM causes the highest conduction loss due to its discontinuous current. The other two have a similar form of current, hence similar conduction loss. The conduction loss increases slightly at lower output voltage.

### 4.2. Switching Losses

The switching losses also play a significant role in the efficiency. Soft switching techniques, including Zero-Current Switching (ZCS) and Zero-Voltage Switching (ZVS), are widely applied to minimize the switching-on loss. If a switch component is turned on while the current is negative (namely through its anti-parallel diode) or zero, ZVS or ZCS is realized [27]. Besides, a snubber (i.e., the drain-source capacitor) is required to reduce the switching-off loss during Miller Plateau. The snubber is supposed to be designed properly according to the gate resistance and switching-off current. On the one hand, a too-small snubber is not able to provide enough current support, and the energy of too large a snubber will cause extra loss at low- or zero-current conditions.

Hard switching not only causes high switching loss on the MOSFET, but it also causes a surge problem on the complementary power diode [24]. If a power diode, with current flowing, is switched off, the reverse recovery effect will result in voltage surge and significant power loss.

The conventional converter has two hard switchings and two ZVSs in each cycle, which leads to a large amount of switching loss. In comparison, the proposed converter with both FFM or VFM achieves soft switching over the entire load range. On the input side, the leading leg ($S_1$ and $S_2$)

operates under ZVS or ZCS condition, thus the snubber is designed for the rated operation point. The lagging leg ($S_3$ and $S_4$) are always turned on and off at zero current. Therefore, this pair of switches requires no snubber [28]. On the output side, all MOSFETs achieve ZCS, which also needs no snubber. If synchronous rectifier is applied on the output side, they can also achieve ZCS.

Although soft switching techniques can largely reduce the switching loss, there are losses during the switching process. It can be approximately calculated from the switching voltage and current by the following equation [29].

$$P_{\text{SW}} = U_{\text{SW}} I_{\text{SW}} t_r \frac{2}{T} \tag{48}$$

where $U_{\text{SW}}$ and $I_{\text{SW}}$ denote the voltage and current at the switch point, respectively. $U_{\text{SW}}$ denotes the DC link voltage. $t_r$ denotes the rising time, $T$ is the operation period. The switching current is the current at $t_1$.

$$I_{\text{SW,FFM}} = i(t_{1,\text{FFM}}) = A_1 \omega_r C sin(\omega_r t_{1,\text{FFM}}) \tag{49}$$

$$I_{\text{SW,VFM}} = i(t_{1,\text{VFM}}) = A_1 \omega_r C sin(\omega_r t_{1,\text{VFM}}) \tag{50}$$

The switching loss under FFM and VFM can be obtained by Equations (51) and (52), respectively.

$$P_{\text{SW,FFM}} = U_1 I_{\text{SW,FFM}} t_r \frac{2}{T_{\text{FFM}}} \tag{51}$$

$$P_{\text{SW,VFM}} = U_1 I_{\text{SW,VFM}} t_r \frac{2}{T_{\text{VFM}}} \tag{52}$$

The switching current of the conventional converter is predicted with FHA. The conventional converter has 1 hard-switching and 1 soft-switching action in each half sequence. Considering the loss on the MOSFET, the complementary diode of a hard switching is twice the loss of a soft switching.

$$I_{\text{SW,CONV}} = \frac{n\pi I_{\text{OUT,CONV}}}{2} sin\left(\frac{U_1 - U_2'}{2U_1}\pi\right) \tag{53}$$

$$P_{\text{SW,CONV}} = 10nU_1 I_{\text{SW,CONV}} t_r \frac{2}{T_{\text{CONV}}} \tag{54}$$

The comparison of switching loss is illustrated in Figure 9.

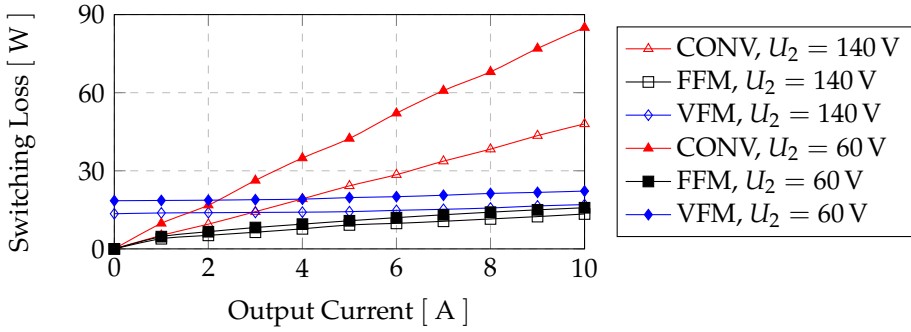

**Figure 9.** Switching loss of three converters ($U_1 = 80\,\text{V}$).

The switching losses of the proposed VFM converter are almost constant, with a small increase. Although the switching current in Equation (50) is small at small output current, the switching frequency in Equation (20) is much higher than the fixed frequency methods. Therefore, compared to the other two modulation methods, VFM causes the highest switching losses at low power. With the increasing current, the switching frequency of VFM decreases, hence the switching losses of VFM and

FFM get close to each other. The switching current of the conventional converter is always the lowest, however, its switching performance is not satisfying due to hard switching.

### 4.3. Total Losses and Efficiency

Under the assumption of ideal inductor and transformer, the total loss is the sum of the conduction loss and the switching loss, as shown in Figures 10 and 11.

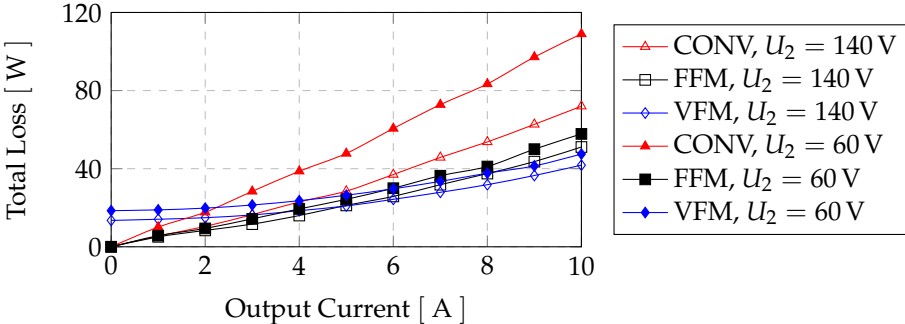

**Figure 10.** Total loss of three converters ($U_1$ = 80 V, $r_{ON}$ = 40 mΩ).

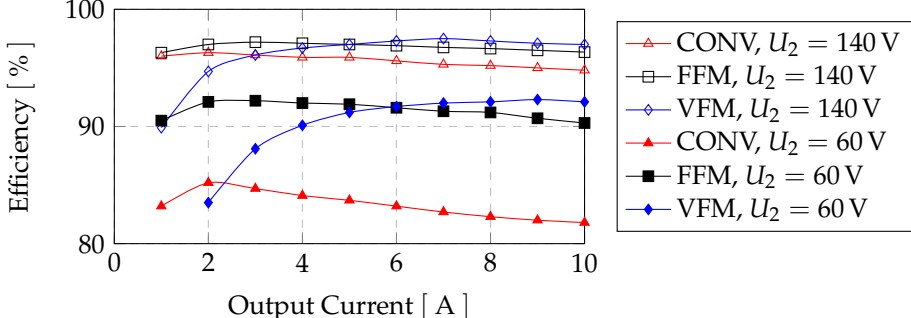

**Figure 11.** Power Efficiency of three converters ($U_1$ = 80 V, $r_{ON}$ = 40 mΩ).

The switching losses play a more important role in light load cases, while the conduction losses dominate the total loss in heavy load cases. The proposed converter with VFM has a greater advantage in a heavy load range, and FFM is more suitable for a light load. With the increase in output voltage from 60 V to 140 V, the benefit range of VFM is widened from $I_{OUT} > 6$ A to $I_{OUT} > 5$ A. In comparison, the conventional converter causes the highest loss due to high switching loss.

The output voltage has a positive effect on the efficiency. The total losses of all three converters will decrease at a higher output voltage, while the delivered power increases in the other case. Therefore, the efficiency is improved by a higher output voltage for both the conventional and the proposed converters. At 60 V output voltage, the proposed converter can achieve over 90% efficiency, while the conventional converter is lower than 85%. In comparison, at 140 V output voltage, the efficiency of all three converters is above 95% in most operation conditions.

### 5. Simulation and Experimental Results

A 1.5 kW prototype of the LC-DAB was built and tested to demonstrate the viability and effectiveness of the proposed topology and modulations (Figure 12). The prototype was built on a printed board by using MOSFET (IRPF4668, 200 V) on the input side and realizes a synchronous rectifier with MOSFET (IPW60R045CPA, 600 V) on the output side. The dead time of each switch pair was set at 2 μs. The controller is coded in DSP (TMS320f28374d, 200 MHz), and the modulation is generated through FPGA (EPM570T100I, 66 MHz). The control board is placed on opposite side of heatsink to minimize the impact of eletromagnetic interference. The ADC on DSP measures the output current, input voltage and output voltage. Then, the DSP calculates the switching frequency and

duty ratio according to the modulation method and the PI controller introduced in Sections 2 and 3. Then, the modulation parameters are sent to FPGA and the FPGA gives the gate signals to the drivers. The designed parameters of the prototype are listed in Table 2.

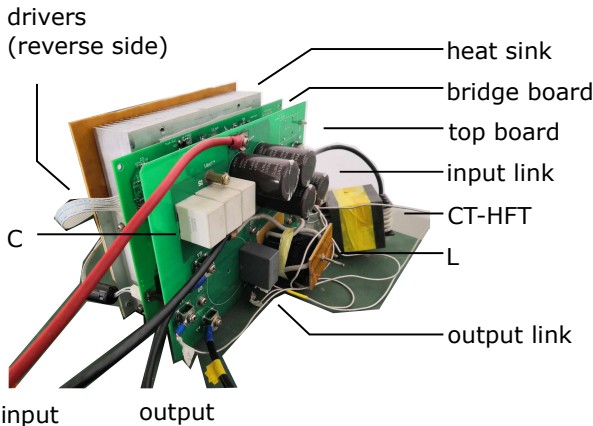

**Figure 12.** Experimental setup.

Figures 13 and 14 illustrate the operation waveforms of the proposed converter with FFM and VFM under different operating points. The simulations are carried out in MATLAB under assumption that all devices are ideal without parasitic parameters. The simulated results can demonstrate the correctness of the theoretical waveform including gate signal sequence, duty ratio caluclation, reverse current blocking, etc., and compare with the experimental results. The simulated waveforms are on the left column and the experimental waveforms are on the right column under the same operation conditions. It shall be noted that the two cases have similar results, consistent with the description in Section 2. Figure 13a–f show the waveforms under FFM. The current returns and remains zero before the next switching action, which operates in DCM. Figure 14 depicts the waveforms under VFM. The results verify that the current is accurately predicted in Sections 3 and 4, and BCM is achieved at the derived turns ratio and frequency.

In the experimental results, voltage oscillation in $u_2$ under FFM can be observed in Stage 3 and Stage 6. In zero-current stages, the current in the main conduction loop remains at zero. In the meantime, the Drain-Source parasitic capacitance of the output side MOSFET and the magnetization inductance of the transformer make up a high-frequency LC series oscillation loop. Therefore, $u_2$ oscillates in zero-current stages, which can be observed in Figure 13b,d,f. If the converter is modulated under VFM, as shown in Figure 14, the oscillation can be avoided due to the elimination of zero-current stages.

The snubber capacitance of $S_1$ and $S_2$ in the proposed converter is chosen as 33 nF in order to reduce channel current while switching-off. For the turning-on process, two ZVS and two ZCS in each sequence of the proposed converter with FFM or VFM can be observed in Figures 13 and 14. The hard switching is avoided, hence the switching loss is reduced. Another advantage of soft switching is reduced overshoot voltages and switching oscillations. There are almost no overshoots and oscillations in the switching process. The elimination of the overshoots can help avoid the damage to the semiconductor due to high voltage and current. The experiment results show that the semiconductors under the worst case (peak power) are safe (maximal 84 V on the input side MOSFETs and 450 V on the output side MOSFETs).

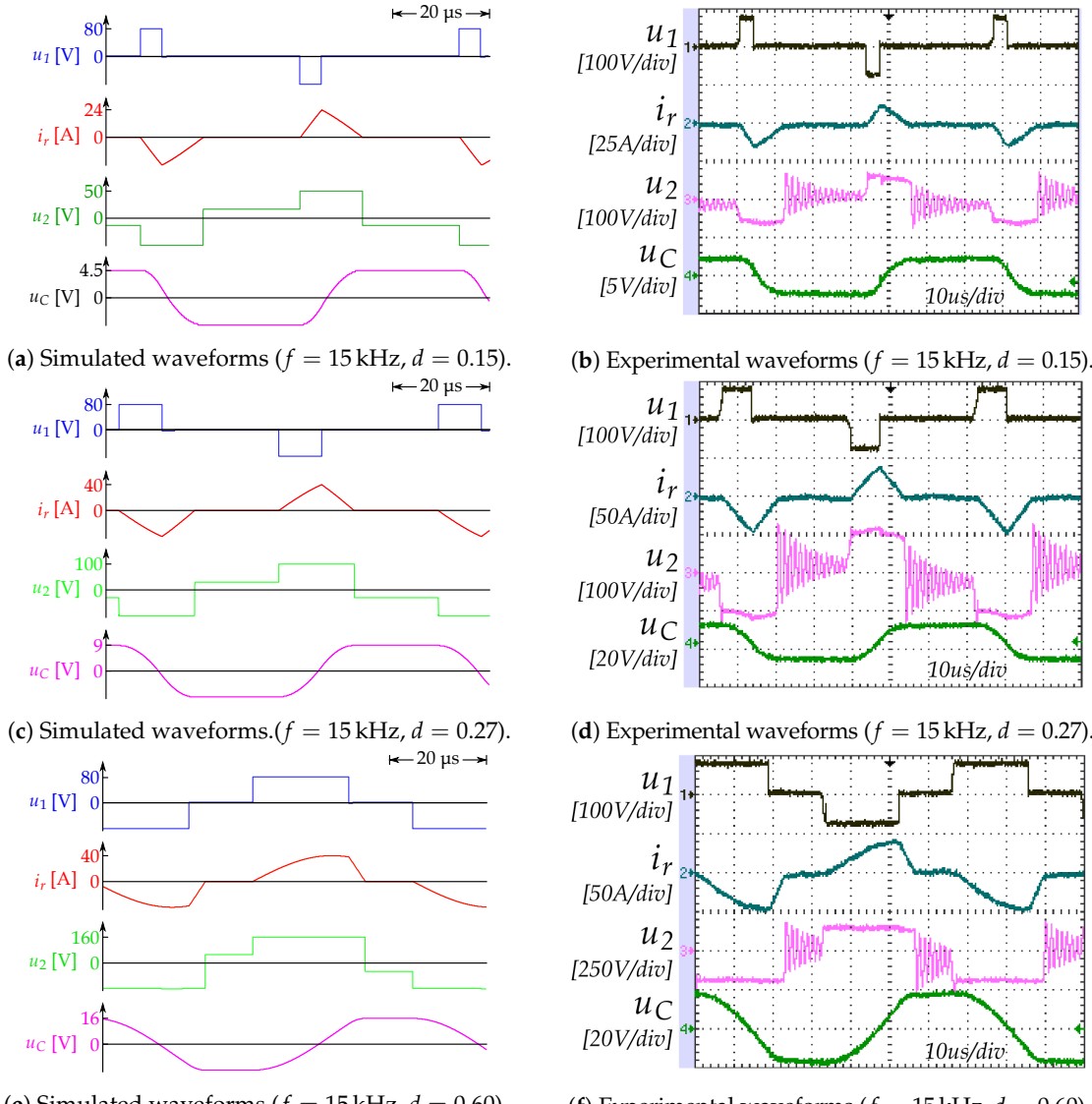

**(a)** Simulated waveforms ($f = 15\,\text{kHz}$, $d = 0.15$).

**(b)** Experimental waveforms ($f = 15\,\text{kHz}$, $d = 0.15$).

**(c)** Simulated waveforms.($f = 15\,\text{kHz}$, $d = 0.27$).

**(d)** Experimental waveforms ($f = 15\,\text{kHz}$, $d = 0.27$).

**(e)** Simulated waveforms ($f = 15\,\text{kHz}$, $d = 0.60$).

**(f)** Experimental waveforms ($f = 15\,\text{kHz}$, $d = 0.60$).

**Figure 13.** Proposed Converter with Fixed Frequency Modulation (FFM) waveforms. (**a,b**) $U_1 = 80\,\text{V}$, $U_2 = 50\,\text{V}$, $I_{\text{OUT}} = 2.5\,\text{A}$. (**c,d**) $U_1 = 80\,\text{V}$, $U_2 = 100\,\text{V}$, $I_{\text{OUT}} = 5\,\text{A}$. (**e,f**) $U_1 = 80\,\text{V}$, $U_2 = 160\,\text{V}$, $I_{\text{OUT}} = 9\,\text{A}$.

The power is measured at the input and output terminals using two sets of Juwei (300 V/100 A) DC power meters, and the curves in Figure 15 show a comparison of theoretical efficiency and experimental efficiency. Both results show the same trend and similar values. FFM achieves higher efficiency at a light load. In comparison, VFM shows better performance in experiments at heavy load, and the power efficiency at rated power 1.5 kW reaches 97.6%, 1.1% higher than FFM.

The power efficiency in the experiment is lower than the theoretical value shown in Figure 15. One possible reason for this is that the internal resistances in the components (inductor, capacitor, CT-HFT and wires) are neglected in theoretical calculation. Besides, the on-state resistance $r_{\text{ON,MOSFET}}$ of the MOSFETs may deviate from its norminal value due to the operating temperature and aging. The difference between the experimental and theoretical results are small, which can demonstrate the correctness of the power loss analysis.

(**a**) Simulated waveforms ($f = 74.1\,\text{kHz}$, $d = 0.429$).

(**b**) Experimental waveforms ($f = 74.1\,\text{kHz}$, $d = 0.429$).

(**c**) Simulated waveforms ($f = 52.6\,\text{kHz}$, $d = 0.632$).

(**d**) Experimental waveforms ($f = 52.6\,\text{kHz}$, $d = 0.632$).

(**e**) Simulated waveforms ($f = 26.3\,\text{kHz}$, $d = 0.816$).

(**f**) Experimental waveforms ($f = 26.3\,kHz$, $d = 0.816$).

**Figure 14.** Proposed Converter with Variable Frequency Modulation (VFM) waveforms. (**a**,**b**) $U_1 = 80\,\text{V}$, $U_2 = 50\,\text{V}$, $I_{\text{OUT}} = 2.5\,\text{A}$. (**c**,**d**) $U_1 = 80\,\text{V}$, $U_2 = 100\,\text{V}$, $I_{\text{OUT}} = 5\,\text{A}$. (**e**,**f**) $U_1 = 80\,\text{V}$, $U_2 = 160\,\text{V}$, $I_{\text{OUT}} = 9\,\text{A}$.

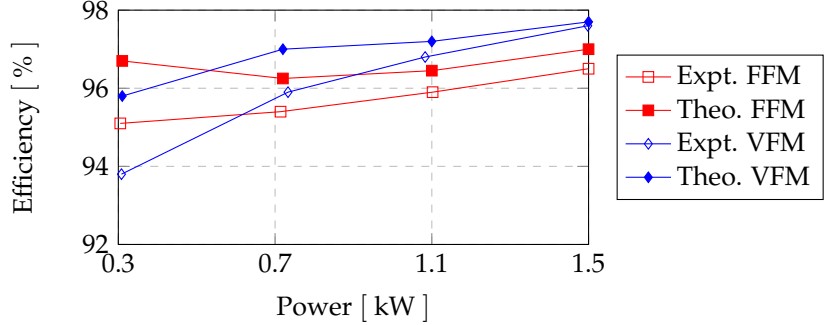

**Figure 15.** Prototype Power Efficiency. ($U_1 = 80\,\text{V}$, $U_2 = 160\,\text{V}$).

## 6. Conclusions

In this paper, a converter with FFM and VFM is proposed. The proposed topology can eliminate backflow power by block reverse current, and the modulation algorithm can accurately predict the current waveform, and realize DCM under FFM or BCM under VFM. The feasibility of the topology and the modulation algorithms are demonstrated through the simulated and the experimental results. A high operating efficiency (up to 97.6%) can be achieved in the 1.5 kW prototype by applying the proposed modulation methods. Furthermore, soft switching is realized and contributes to the safety of the semiconductors. By appplying this well-performed DC-DC converter in situations such as solar energy generation, electric vehicle charging and DC smart grid, the losses of energy can be reduced and the switching devices' safety can be improved.

In future studies, the following limitations of the proposed converter can be addressed. The modulation method is only designed for the condition $U_1 > U_2'$. VFM is not suitable for a multi-converter system, and finally, the 3-winding center-tapped transformer will increase the size of the converter.

**Author Contributions:** Conceptualization, G.C. and M.M.K.; methodology, G.C.; software, N.X.; validation, G.C. and N.X.; formal analysis, G.C. and M.M.K.; investigation, G.C.; resources, L.Y.; data curation, G.C.; Writing—Original draft preparation, G.C. and M.H.; Writing—Review and editing, G.C. and M.M.K.; visualization, G.C.; supervision, M.M.K.; project administration, L.Y. All authors have read and agreed to the published version of the manuscript.

**Funding:** This research received no external funding.

**Conflicts of Interest:** The authors declare no conflict of interest.

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
