# Peer review of "A DC-DC Center-Tapped Resonant Dual-Active Bridge with Two Modulation Techniques"

_electronics, doi:10.3390/electronics9101699_

Round 1

Reviewer 1 Report

In this paper, an enter-tapped LC series resonant dual active bridge (LC-DAB) converter for DC-DC conversion is proposed is presented. The feasibility of the topology and the modulation algorithms are demonstrated through the theoretical and the experimental results. The modulation algorithm realizes discontinuous conduction mode under fixed frequency modulation or border conduction mode under variable frequency modulation. The proposed modulation techniques guarantee soft switching for all devices. The synchronous rectifier is realized by the center-tapped bridge to reduce the conduction losses.

The paper is well organized and carefully developed. Nevertheless one can find some errors:

  1. Page 1, line 30: inappropriate reference (The system of center-tapped bridge is not described in the mentioned reference).
  2. Page 19, line 434: The name of the first author of the article has been changed.
  3. Equation (1): In the numerator of the formula (1) there should be the number 1, not 2.
  4. Equation (18): The variable t1 cannot be in the denominator of the formula.
  5. Equation (20): The variable t1 cannot be in the denominator of the formula.

Author Response

Dear reviewer,

Special thanks to you for your good comments.

Kindly see the attachment for our responses.

best regards,

Gengxin

Reviewer 2 Report

In this paper, the authors proposed a center-tapped LC series resonant dual active bridge (LC-DAB) converter for DC-DC conversion. The key ideas of the proposed technique are modulation methods, namely fixed frequency modulation (FFM) and variable frequency modulation (VFM). Overall, the authors have made a good attempt, I think. However, due to the lack of comparison data based on simulations/experiments, the effectiveness of the proposed technique is not clear. The authors must justify the effectiveness of the proposed technique by comparing with existing methods. The reviewer’s comments are as follows:

Reviewer’s comments:

  1. The research survey is not enough. The articles quoted in Sect. 1 are not state-of-the-art. (Of course, Refs. [1] and [2] are new articles. However, these references have no relationship with the proposed technique.) The authors should survey past studies in detail.

  1. The authors illustrate the proposed converter topology in Fig. 1. However, the novelty of the proposed topology is not clear. The topology of the proposed converter is almost the same of that of the traditional dual active bridge (DAB) converter. The authors must describe the novelty of the proposed topology.

  1. In Table 1, please quote articles for “Con. LC-DAB”.

  1. In the theoretical analysis, the authors should clearly describe the assumption. For example, the authors ignored some important factors such as on-resistance of switches and threshold voltage drop cause by diodes.

  1. How did you obtain the simulated result shown in Fig. 6. Please explain the simulation conditions.

  1. In Sect. 4, the authors compared the proposed converter with the existing converters proposed in [13, 14, 16]. However, Refs. [13, 14, 16] do not have the center-tapped DAB structure. The authors demonstrate the discussion between the proposed converter with the existing converter with the center-tapped DAB structure.

  1. In Sect. 5, the simulation conditions are not clear. How did you determine the device model (physical model) of the switches IRPF4668 and IPW60R045CPA? As you know, the device model of circuit components strongly affects the simulated characteristics. Please explain it clearly. Besides, what kind of simulations did you perform?

  1. In Sect. 5, the circuit components for the experimental circuit are not clear. For example, how did you realize the controller? Please explain it.

  1. The effectiveness of this work is not clear. Through simulations/experiments, the authors must justify the effectiveness of the proposed converter by comparing with the existing converters. Several articles are listed in references. However, no comparison is shown with these converters in Sect. 5. The authors only demonstrated the theoretical data in Table 11. Please show comparison data with existing converters.

  1. In Fig. 15, please explain the reason why there is errors between the experimental results and simulated results.

  1. In the introduction part, the authors described that “Hence, the modulation techniques not only reduce the converter power losses for the entire load range, but also the costs.” However, this interpretation is not supported by any demonstrations in Sect. 5. e.g. There is no discussion about cost, such as component counts. The authors should demonstrate it clearly.

  1. The results of this research are not clear in Conclusions. Show the scientific contribution of this work with concrete data.

  1. To help readers’ understanding, the applications of the proposed converter should be described. What’s the final goal of this research?

Author Response

(The authors gave the same response as above.)

Round 2

Reviewer 2 Report

[Accept]

Title: A DC-DC Center-tapped Resonant Dual Active Bridge with Two Modulation Techniques

In this paper, the authors proposed a center-tapped LC series resonant dual active bridge (LC-DAB) converter for DC-DC conversion. The key ideas of the proposed technique are modulation methods, namely fixed frequency modulation (FFM) and variable frequency modulation (VFM). In the first version, the effectiveness of the proposed technique was not clear due to the lack of comparison data based on simulations/experiments. However, in the revised version, the weak point of the first version was significantly improved. The revised version is well written and organized paper, I think. It is scientifically sound and contains sufficient interest to merit publication. So, I’d like to accept this paper.